# The Significance of Taking Antiretroviral Medications for African-Born People Living with HIV and Residing in Minnesota

**DOI:** 10.3390/pharmacy8020108

**Published:** 2020-06-26

**Authors:** Alina Cernasev, William L. Larson, Cynthia Peden-McAlpine, Todd Rockwood, Paul L. Ranelli, Olihe Okoro, Jon C. Schommer

**Affiliations:** 1College of Pharmacy, The University of Tennessee Health Science Center, 301 S Perimeter Park Drive, Suite 220, Nashville, TN 37211, USA; 2Allina Health Uptown Clinic, 1221 West Lake St., Suite 201, Minneapolis, MN 55455, USA; William.L.Larson@allina.com; 3School of Nursing, University of Minnesota, 308 Harvard Street SE, Minneapolis, MN 55455, USA; peden001@umn.edu; 4School of Public Health, University of Minnesota, 420 Delaware Street SE, Minneapolis, MN 55455, USA; rockw001@umn.edu; 5College of Pharmacy, University of Minnesota, 232 Life Science Duluth, 111 Kirby Drive, MN 55812, USA; pranelli@d.umn.edu (P.L.R.); ookoro@d.umn.edu (O.O.); 6College of Pharmacy, University of Minnesota, 308 Harvard Street SE, Minneapolis, MN 55455, USA; schom010@umn.edu

**Keywords:** HIV/AIDS, medication experiences, African-born persons living with HIV

## Abstract

Thanks to the development of antiretroviral (ART) medications, HIV is now a chronic and manageable disease. This study aimed to (1) capture the experiences of African-born persons living with HIV and taking antiretroviral treatment, and (2) explore the impact of social and cultural factors on their decisions to follow the prescribed treatment. For this study, a qualitative approach was used. The participants were recruited via fliers, then screened for inclusion and exclusion criteria. Recruitment of the participants continued until data saturation occurred. The interview guide was developed based on the extensive literature and recommendations from the clinical team. In-person narrative interviews were conducted with 14 participants—African-born persons living with HIV and residing in Minnesota. Thematic Analysis revealed three emergent themes: “*To exist I have to take the medicine*”; barriers and facilitators in taking ART medications; and the power of spirituality and prayers. The findings of this study paint a picture of African-born persons living with HIV, and their experiences with ART medications. This study not only presents the participants’ medication experiences and their significance, but also tells their stories of how God and prayers play a significant role in helping them to get through the difficult moments of their lives.

## 1. Introduction

Initially, in the early stages of the HIV pandemic, the African reports did not receive much attention. It took nearly a decade for scientists and clinical staff to recognize the pandemic on the African continent [1]. The recognition resulted in different resources being made available to Africa. Different countries in Africa have been affected differently by the HIV pandemic. For example, in the early 1990s, in countries that belonged to the “AIDS belt,” nearly 25% of the people living in the cities were suffering from HIV [2]. A recent 2017 World Health Organization (WHO) report estimated out of 1.8 million global new infections, 1.2 million occurred in Africa [3]. 

On the other hand, in the United States (U.S.), there are more than 1.1 million people living with HIV (PLWH) [4]. The diagnoses have had a disproportionate effect on different minorities, such as African Americans. For instance, most of the HIV diagnoses occurred in the southern states. In 2016, 44% of the new virus diagnoses were in African Americans, who represent 12% of the U.S. population [4]. The Center for Disease Control (CDC) reports the statistics for African Americans; however, the reports do not differentiate between African-born and African Americans. Therefore, there are no nationwide specifics to differentiate the number of African-born persons living with HIV in the U.S.

African-born PLWH have been disproportionately affected by HIV in Minnesota. Data showed a steady rise in new HIV infections among African-born PLWH in Minnesota in the last decade [5]. To illustrate, there were eight new HIV infections in 1990, while in 2002, the infections increased to 65 new HIV infections [6]. Although white, non-Hispanic people represent the highest number of PLWH in Minnesota (4117 cases in 2018), they have one of the lowest rates of PLWH (91 per 100,000 persons) [7]. Furthermore, the State recorded a total of 1468 African-born PLWH in Minnesota in 2018 [7]. Consequently, the rate of African-born PLWH is 1400 per 100,000 persons, more than 15 times greater than that of white people and non-Hispanic people [7]. The African-born population makes up 2% of Minnesota’s population, based on the 2010 US Census data. The recent report does not specify the number of African-born PLWH who receive antiretroviral (ART) medications.

The first hope for HIV treatment occurred in 1987 when zidovudine, also known as AZT, was launched on the market [8]. In the following years, other medications for the treatment of HIV received approval for human use. For example, in 1995 there were only five antiretrovirals, while in 2000 the market saw a sharp increase to nearly 16 medications [9]. Furthermore, in 2018, at least 40 ART medications emerged on the U.S. market, in five different classes to select from for treatment [10]. The main advantages of the newer medications are that they are less toxic, more tolerable than the initial ones, available in combinations, and dosed once daily [11,12]. 

The development of ART medications for HIV over the last two decades has transformed treatment for HIV, from an acute, palliative focus, to a long-term, managed effort. According to the current guidelines, all HIV positive persons should be started on ART regimen regardless of their CD4 counts [13]. The rationale for this recommendation is to decrease mortality and morbidity associated with HIV infection, and reduce transmission [13]. Commonly, an initial regimen consists of three ART medications, that belong to at least two different classes [13]. This is changing with the introduction of two drug combinations, with more in development [14]. 

The importance of ART adherence was described by various authors [15,16]. Patients should take 95% of the prescribed doses to obtain viral suppression and avoid resistance [17]. Maintaining adherence to the ART medications, and achieving viral load suppression, eliminates the risk of virus transmission [18]. In other words, a patient with undetectable HIV viral load does not transmit the virus [19]. Both the CDC and Minnesota Department of Public Health publicly endorsed the campaign of "U = U" or "Undetectable = Untransmittable" in October 2017. The goal of this public campaign is to promote that undetectable viral load achieved through ART means unable to transmit HIV to sexual partners. [20]. 

The objectives of this study were to (1) capture the lived experiences of African-born people living with HIV who are taking ART treatment, and (2) explore the impact of social and cultural factors on their decisions to follow ART treatment.

## 2. Methods

A qualitative approach was used for this study. Storytelling is also known as narrative interviews, and these two terms are used interchangeably in the literature. A vital aspect of the narrative interview is that it elicits the story of individuals. The narratives invite all the participants, including listeners, readers and viewers, to enter the perception of the narrator [21]. Storytelling uses an unstructured interview with probing questions if the participant does not discuss the topic in the probes [21,22]. 

### 2.1. Participants and Data Collection

Participants learned about the study through fliers placed in HIV clinics, pharmacies, and service organizations that served HIV-infected people, located in the metropolitan area of Minnesota. Fliers described the nature of the study and provided a telephone number for participants to call. Participants who called were screened for eligibility.

The study entry criteria were as follows: (a) 18 years of age or older, (b) spoke English, (c) born in Africa, (d) HIV-positive, (e) domiciliated in Minnesota, (f) prescribed ART treatment, and willing to share their experiences of being HIV-positive and receiving ART treatment. The exclusion criteria were as follows: (a) minors, (b) did not speak English, (c) were not born in Africa, and (d) were HIV positive, but did not take ART treatment at the time of the interview.

Qualitative studies do not rely on large sample sizes to claim validity for the concepts generated. Sample size in qualitative studies has a different meaning than in quantitative studies, and does not rely on power [23,24]. A target sample size demands 10 participants [24] or simply enough for detailed descriptions and rich themes to emerge from the interviews, therefore reaching a level of saturation [25]. Saturation has been defined as a point beyond which no significantly new information is obtained [25]. Recruitment of the participants continued until saturation occurred [26].

Semi-structured, in-person interviews were conducted with the recruited participants at the time and location selected by the participant. The interview questions were developed by the principal investigator based on extensive review of the literature (See Appendix A). To ensure that the instrument aligned with the scope of this research, the principal investigator consulted with her team comprised of experienced researchers and health professionals (pharmacists and physicians), who specialize in HIV/AIDS. In the fall of 2017, a pilot interview was conducted with an African-born participant who was not HIV positive. The interview was not audio recorded. Additionally, the questions were tested on three participants who were African-born and for whom English was not their mother tongue. The interview questions pertinent to these objectives are presented in Appendix A. A more detailed description of this methodology has been published [27]. While the previous manuscript focused on addressing the stigma associated with HIV, the current paper captured the experiences of African-born PLWH who were taking ART treatment.

Approval of this study was received from the Institutional Review Board (IRB) of the University of Minnesota (STUDY00001597). The study was initiated in December 2018 and recruitment occurred until April 2019, when saturation of data occurred. Informed consent was obtained from each participant prior to the interview. If names were mentioned unintentionally during the interview, they were blocked out in the transcripts.

### 2.2. Data Analysis and Rigor of Data

Interviews of all 14 participants were audio recorded, and the records were transcribed verbatim by a commercial service (QHR Consultants, Madison, WI) to avoid any bias. The team ceased recruitment of the participants once saturation was achieved [28]. The interview transcript was analyzed using Thematic Analysis, which enables the themes to arise from the data [29]. This method has been described in qualitative analysis as inductive coding [29]. The inductive codes were combined, based on the similarities, into categories that led to the emergence of themes [29]. During the inductive coding, the principal investigator wrote analytic memos that facilitated the abstraction of the meaning from the data [30]. The coded text was condensed into categories that emerged into themes [29]. The data analysis was completed in Dedoose, a qualitative software.

Rigor in the study was achieved by using the framework proposed by Lincoln and Guba [31]. For example, confirmability was obtained by using a three-member research team who reviewed the codes. One researcher reviewed the codes of two transcripts, while the other researcher coded four interviews independently. Then the team met several times to discuss the codes, and consensus was achieved. Credibility and dependability were obtained throughout the data collection and data analysis. In addition to the analytical memos, the principal investigator wrote summaries that captured non-verbal communication that might be relevant to the data analysis. These summaries also included the principal investigator’s thoughts on the interview questions that might need to be rephrased, and areas for further questioning when interviewing other participants. Transferability was addressed by providing dense descriptions in the findings that reflect the lives of PLWH.

## 3. Results

A total of 14 participants were interviewed. Eight participants were female and six were male. Table 1 highlights the demographic information, the year and country of the diagnosis, and the country where treatment started.

Three themes emerged from the analyzed data that relate to the main objectives. In the first theme, the participants share with us their medication experiences, and the significance of taking ART medications. In the second theme, their stories explore the factors that promote or inhibit taking ART medications. In the last theme, it is shown that God and prayers play a major role in helping the participants to get through the difficult moments of their lives.

### 3.1. Theme 1: “To Exist I Have to Take the Medicine”

A key characteristic of the narratives was that the participants take the prescribed medications “to be alive”, and to spend time with family. Regardless of where they began treatment, these participants presented similarities, including the experience of side-effects, “pill burden,” symptoms that prompted them to start ART treatment, the size of medications, and secrecy about taking medications.

For Participant 10, the desire to live in a time when there will be medication to cure the disease is her driving force to “be alive.” In the following quotation, Participant 10’s use of the term “I want to live!” twice suggests a sense of a relationship between her, the ART medications, and her life. Clearly, Participant 10 transmits an important message that “she wants to live!”


*“I want to live! (chuckles) I want to live! I want to be there to tell the story. I want to be there to see the [grandchildren] you know the cure, to be there. And we can be the pioneers to say you know we were there then, and now the medication is here. I want to see that. That’s what I want to see. Yes! That’s what drives me…”*


The ART medications helped Participant 4 achieve a regular life and, ultimately, sustain it. He elaborates:


*“…taking medication is, ya know, it help a lot because I remember before I started my medication, I was really sick, ya know, continuously sick today, sick tomorrow, sick another day. But until I started my medication now, ya know I’m like, normal person and yea…Never been sick like, ya know, that I was before- before I start the medication…”*


Several participants (*n* = 9) were started on ART medications at least a decade ago. Some of them noticed the differences between the size of the tablets or even the “pill burden.” For Participant 10, the current regimen is much more appealing to take, since the medications are smaller in size compared to the previous ones, that were much larger and more difficult to swallow. The participant uses the term “tiny” repetitively, suggesting an emotional reaction to the regimen.


*“… back in Kenya the medication, what I used to take was very thick and very big (Laughs) and you know just swallowing that like every day, it was hard. But here they, very tiny! Very, very, tiny medication, very tiny medication so, yeah, it’s easy to take…”*


All the narratives are distinguished by secrecy, even though decades had passed since they were diagnosed with the disease and started taking ART medications. A few of the participants highlighted the emotional challenges of keeping the medication a “secret.” Even though taking ART medications was vital for the participants, one participant recollects how she used a private space to take them.


*“Taking medication, I would sneak to go hide by the toilet to take my medication (chuckles) because I didn’t want anyone to see. Yeah.”*
(Participant 10)

Similarly, Participant 13 experienced the emotional challenge of how to keep the medication a “secret.” She asked the medical team to use a different package for the medications she was taking. This camouflage allowed her to continue taking the prescribed regimen, and not share her “secret” with her children until she was ready emotionally to talk to them.


*“But I used to hide them so well. The first time when I was started on medication, she {her daughter} was in the house. I told the doctor… Can you repack my medication because the girl was in the house? I have two boys and one girl. So, they repacked.”*


Some participants (*n* = 7) described experiencing different side effects, that occurred at different stages during treatment. Even though the participants described experiencing various side effects, they continued taking ART medications. Depending on the circumstances, the drug that produced the side effect might have been changed or discontinued. However, no common side effects emerged from the interviews.

### 3.2. Theme 2: Barriers and Facilitators in Taking ART Medications

This theme describes the factors that promote or inhibit adherence to ART medications in the interviewed participants. Similar to the previous theme, where the participants took medications to remain alive, the primary purpose for being adherent to the treatment was to continue their lives. This theme is divided into two sub-themes: (a) facilitators, which promote adherence, and (b) perceived barriers to adherence.

#### 3.2.1. Facilitators that Promote Adherence to Medications

Some of the participants (*n* = 8) use various tools, such as a pillbox or alarm, to ensure they are adherent to the prescribed regimen, while others do not use them because their medications are part of their routine.

The pillbox was mentioned as an adherence tool by many interviewed participants. The following quotation describes the significance of the pillbox in Participant 8′s daily life. He highlights the differences in his symptoms between when he was not adherent, and now that he is adherent. Therefore, the pillbox plays a dual role in his life: it helps him to remember to take the medications, and improves his adherence to the prescribed regimen. He states:


*“I used to But not anymore. When I miss it, you know I get other stuff, and just miss it and I get sick, cold, cough and any other thing. So, {pillbox} for seven days. I set it up. That’s why I don’t miss it Monday, Tuesday morning, or in the evening.”*


Taking medications daily for the rest of their lives was a new concept for many of the participants. Furthermore, it was initially a significant challenge for many of the participants to remember to take the prescribed medications. Again, we can see in the extract below the importance of using a pillbox. The pillbox helped Participant 14 to develop a routine. These days, the medications have become a part of his life, and he does not need the pillbox as a reminder.


*“Um- First of all at the beginning it was very hard. Because I’m not used to taking medications, and sometimes I forget to take medications. And I remember they gave me one of these-{pillbox}. Yeah, so I put all the pills in the boxes to remember better. After a few years it become almost a part of me. You know? A habit. It’s automatic. Yeah, nobody need to remind me or stuff like that. So, I don’t miss my pills.”*


Similar to the previous participant who does not depend on the pillbox, Participant 9 does not use the alarm these days as a reminder. This extract demonstrates how medications became part of her life. She says:


*“I used to have my alarm on my phone that, by exactly 10 min to 10, the alarm would sound but then not anymore because once you’re used to doing something it becomes a routine, so definitely when it reaches that 10, I already know it’s time for me to take my medication.”*


#### 3.2.2. Perceived Barriers to Adherence

A few participants discussed some constraints that could interfere with adherence to the medications, including traveling to another city or staying with a family friend for an evening. Three participants stated that they were not able to adhere to the medication schedules due to some insurance issues.

The problem for Participant 4 is that he forgot to pack the ART medications when traveling.


*“This is my little problem sometimes, ya know, when I went out, like going to another city. I used to forget, sometime…”*


For Participant 4, the change in insurance at the beginning of the year prevented him from taking the prescribed ART medications. According to his narrative, the pharmacy was not aware of these insurance changes, which prevented the participant from taking the medications for nearly a week. Fortunately, he was able to solve the issue by calling the case manager, and the insurance was reinstated. The repetitive use of words “I had to stay without medication” in this excerpt highlights his willingness to take medications, while being unable.


*“Uh, it was this year, I stayed like, more than four days without taking my medication. It happened that when I switched the- the insurance... And I- I already ran out of medication… And I have to stay, ya know, without my medication… And that one it took a while and I’d say like almost four to five days without medication.”*


On the contrary, Participant 12 was not able to solve the insurance issue for nearly six months. He was dropped due to some errors from the subsidized government insurance plan. Not having insurance coverage resulted in not receiving his necessary medication regimen. The CD4 counts decreased, the viral load rose, and he was more susceptible to opportunistic infections. Furthermore, the increased viral load meant it was not undetectable, and he could transmit the virus. In the quote below, the participant discusses how this insurance lapse affected his life.


*“…at one point I was out of medication for almost like 6 months. Why? The papers were not going through and then they put be back on and like that, when you’re out of medications the viral level increase and you got sick and you get back in the hospital again so. I went back on the insurance and then got back on my regimen and it’s like you’ve got to start all over again...”*


### 3.3. Theme 3: The Power of Spirituality and Prayers

The participants’ strong beliefs in God offered them strength and support in stressful situations, such as diagnosis and living with HIV. According to their statements, God continuously plays an essential role in their lives. The narratives highlighted the healing powers of prayer, and its vital impact on their daily lives.

Participant 7 emphasized her strong belief in God, and how daily prayers allowed her not to take ART medications for a decade. In this extract we can see a strong bond between her and God.


*“Do you know I stayed 10 years without taking {ART} medicine? From 1994 I come to take medicine 2004. The whole time is faith, when I get sick, I would call my pastor, we would pray and I think yeah, the belief, only the faith, that faith just to believe, it helped me so much, even today. That is how I don’t think medicine only, the medicine also I have with God.”*


In the quotation below, we could sense how the healing powers of prayer transformed the participant. This transformation was observed by her friends who could see it on her face. She says:


*“Yea. Every time I pray, I said, even my friends, people from abroad, they would see me- I mean, my pictures, ‘Why you not working? Why you look better than some people that working?’ I said, ‘Well, it’s God. It’s just by the grace of God I’m living.’”*
(Participant 3)

Some of the participants (*n* = 7) described religious activities they were taking part in, such as attending Mass, engaging in church events, or studying the Bible together. The participants did not want the religious community to know about their HIV positive status, because the disease is stigmatized. Even though some of the participants belonged to a religious community, they would be very careful about to whom they disclosed their HIV positive status. Furthermore, if they shared their HIV status in the religious community, it would be with a person they trusted, and in most cases, they shared the diagnosis with another HIV positive person, who could understand and would not judge them.

Participant 13 discusses Christian life, filled with hope, support and love for each other. The participant discusses the help she received from her church to go through difficult periods of her life. She established a connection with God and used this connection to facilitate her wellness. The participant discusses her Christian faith, such as going to church, praying, supporting those in need and believing in God’s love. Note the frequency of her use of the word “God” or “He” in the following quote. It is clear this person believes in the miraculous powers of God, who is the center of her universe.


*“My driving force is that I’m a Christian. We believe in life after, and in our church also we teach health… We support people with HIV but some people there’s still stigma… We understand that when your day comes to die, is when God has decided that day for you... All this I’ve gone through; God has been there with me. I’m still walking with Him and I will live my full life until when He said I’ll die is when I’ll die…”*


On the contrary, another participant feels he does not have to go to church to pray. Indeed, he developed a direct relationship with God via the Bible and prayer. Participant 8 points out that his family isolated him because of his sexual orientation and diagnosis. The extract suggests that prayers have a healing effect on him, as he elaborates:


*“Oh, even though I’m no sinner, I’m not perfect, I believe in God. Deeply. I listen to Gospel, music. This is keep me go- keep me everything- spiritual. So, it’s the Bible which- Keep me going. Even though I’m gay. This is- It’s what- Kept me going. I feel sad I cry, I’m not accepted, you know by, my own family my siblings, because I’m gay and a lot of things happen to me…”*


The profound connection with Divinity made a few participants change their sexual behavior. Participant 10 emphasized that abstinence from sexual intercourse was due to her Christian faith. She says:


*“…Um, I think being a Christian and having that faith and knowing that life is precious and um- of course for me after I got- after I knew my status then I stopped like, like having multiple partners because I knew, OK, one- my Christian faith wouldn’t allow it. Like it was wrong to, to go around or whatever…So, I stopped you know... having those Christian values in me made me I would say made me healthier…”*


## 4. Discussion

The findings of this narrative study illustrate that, regardless of the country of origin, the participants’ main reason for taking ART medications was “to be alive”, and to spend time with the family.

Several of the participants noticed an improvement in their medication regimen since starting on ART medications a few decades ago. While some of the participants described a decrease in the number of ART medications taken daily, others mentioned a reduction in the tablet size, or even the side effect profile. However, the large tablet size or the pill burden did not prevent any of the participants from taking the prescribed medications. Earlier research conducted with African-born HIV-positive women living in London described the process of taking ART medications as “invasive, disturbing, and disgusting.” [32] (p.2155). These findings are contrary to ours, in which none of the narratives describe the act of taking ART medications as aggressive or revolting. Furthermore, the authors suggested further studies to ask the study participants about their “feelings around the physical futures of the tablets.” [32] (p.2159). Our current study asked each participant about the physical characteristics of the tablets, and most of the participants did not comment. One possible interpretation of these findings was answered by one of the participants who compared her ART regimen in Africa to that in the U.S. According to her statement, “the tablets are tiny and easy to swallow in the U.S.”

Several of the participants described the process of hiding their ART medications. A previous study conducted in London also noted that some of their participants kept their medications secret [32]. The findings of the current study have to be discussed in the social context, where the fear of being stigmatized by family and colleagues played a major role in them hiding the medications. Although a few of the participants were started on ART medications in the U.S., they would not disclose their medication regimens to immediate blood relatives, including their children. To avoid HIV stigma in some South African communities, the patients might use various practices, including hiding the ART medications or crushing the tablets [33]. Although the participants moved out of their country of origin, a few of the participants still tended to hide their medications. Therefore, our novel findings are suggestive, and should be taken into consideration when pharmacists interact with this population.

Our data showed that the prescribed ART medications play an important role in the participants’ lives, and all of them wanted to be adherent to medications to continue living their lives. Furthermore, the participants perceived their adherence to the prescribed ART medications as being very high. Earlier research showed similar results, where the participants’ main reason for having high rates of adherence to ART medications was to remain healthy [34]. On the contrary, a quantitative study conducted in Minnesota within the African-born population showed a lower adherence rate to ART medications, and the study did not provide insights regarding the reasons for adherence [35]. 

A few participants mentioned that another vital reason for being adherent to ART medications was to avoid transmission of the virus. These unique findings are suggestive, and show a connection between the participant and the reasons for being adherent. It is imperative for PLWH to obtain a sustained adherence to ART and achieve viral load suppression in order to reduce the risk of viral transmission [36]. When the medical team explains the importance of ART adherence, terminology such as “viral load suppression”, “reducing the transmission of the virus”, or “not transmitting the virus to another person” is commonly used. Consequently, a few of the interviewed participants described this critical concept, which is related to achieving viral load suppression without being prompted by the interviewee.

Most of the interviewed participants mentioned the usage of various adherence tools, such as a medication organizer, phone alarm or alarm watch. These adherence tools not only help them to be adherent to the ART medications, but also to sustain their daily lives with loved ones. Our study findings corroborate prior research indicating that these types of adherence tools provide support and increase adherence to the prescribed medications [37]. Furthermore, because this study focuses on the participants’ perspectives regarding the importance of adherence, and not the actual rate of adherence to the ART regimen, the self-presentation bias is not a genuine concern.

A few of the narratives discussed insurance issues, such as loss of coverage or lack of insurance coverage for a period of time, as the main impediment to adhere to the ART medications. Our findings bring more evidence to the existing literature that showed that PLWH who had no insurance coverage were at an increased risk of not taking ART medications [38]. Despite the fact that the participants had insurance coverage, a number of them indicated that the co-pay might represent a financial burden.

Although previous studies have shown the role of religion in the lives of PLWH, a novel finding of this study is the role played by a monotheistic religion in all the participants’ lives, regardless of their country of origin and their faith. The two major religions that predominate on the continent of Africa are Christianity and Islam. These religions are monotheistic; thus, they pray to only one God. The participants in this study explained that their belief in God provided them the power and strength to endure the difficult moments in their lives. Consequently, our present study identified common factors, such as the power of prayer and the role of God, across seven different African countries.

The influence of Divinity and Christianity on PLWH has been highlighted by various studies conducted in Africa [39,40]. The profound influence of religion could be observed in a few narratives representing different African countries, where the participants reported a change in their sexual behavior not only due to the disease, but also due to the conservative Christian belief they embraced.

Findings from national surveys conducted globally reported which African countries are predominantly Christian or Muslim. According to this report, Christianity predominated in some countries: Zambia (98%), Kenya (85%), Liberia (86%), Ethiopia (63%), Tanzania (61%) and Togo (44%). Guinea (11%) is reported as predominantly Muslim in this report [41]. Within our data, it could be said that all the participants reported the strong impact of religion, and especially Christianity. This could be attributed to their Christian origin, wherein religion plays a dominant role in their lives [41]. 

References to Christian-related activities were present in most of the narratives. While some of the stories demonstrated their engagement in church activities, including the organization of Bible studies, others mentioned the organization of support groups for people with HIV. Prince, Denis and Van Dijk [42] have also emphasized the growing impact of the Christian religion on shaping the responses to HIV/AIDS epidemics in various countries in Africa. The authors highlighted the increasing number of religious organizations and their involvement in fighting HIV in Africa [42]. Our data suggest that there are opportunities for PLWH to engage in church-related activities that fight HIV stigma on all fronts.

### Strengths and Limitations

This study uncovered unique findings by presenting detailed stories of African-born participants’ experiences with ART medications. The narrative interviewing methodology enabled us to capture these stories and interpret results in context. Additionally, the participants in this study represented seven countries in Africa, and their medication experiences may vary within their own country and culture. Therefore, this cultural diversity among our participants enhances our understanding of how cultures shape African-born PLWH’s medication experiences.

These study results should be considered in light of some limitations. As mentioned above, all of the participants in this study reported themselves as adherent to the ART medications. The perceived high adherence rate in this study could not be checked against data claims or medical records. Nevertheless, even within this sample, all of the participants had a form of insurance at the time of the interview. Finding non-adherent or non-insured participants was challenging because recruitment occurred via fliers through pharmacies and HIV clinics. Consequently, this sample might not be representative of the full spectrum of the African-born PLWH in Minnesota, and may limit the generalization of the findings to the wider population of African- born PLWH in Minnesota.

## 5. Conclusions and Future Studies

ART medication is essential for PLWH to stay alive. It is important to highlight that although most of the participants in this study were started on ART medications when the disease was a “death sentence,” the ART regimen helped them to continue life. Additionally, the study results reveal that participants’ rationale for taking the medication regimen was to achieve and maintain a suppressed viral load, so as to not transmit the disease to another person. Participants used various strategies to remain adherent to the ART medications.

The participants were still learning how to navigate the unfamiliar U.S. healthcare system. Insurance posed one of the main barriers to ART access in this study, rendering the medications too expensive and thus preventing patients from obtaining them. Therefore, it is recommended for pharmacists to be cognizant of social and cultural norms, by asking patients more about their health insurance and financial situations in order to fully understand their specific adherence problems.

Furthermore, the study indicates that the participants believed in a monotheistic God who played an essential role in their daily lives.

Future research should consider these findings in studying how to better serve and care for this patient population.

## Figures and Tables

**Table 1 pharmacy-08-00108-t001:** Demographics of the study participants.

Participant	Gender	Marital Status	Country of Origin	Country/Year of Diagnosis	Country Where Treatment Started
1	Female	Not disclosed	Tanzania	U.S.	U.S.
2	Female	Widow	Zambia	U.S./1996	U.S.
3	Female	Divorced	Liberia	U.S./2008	U.S.
4	Male	Married	Ethiopia	Ethiopia/2006	Ethiopia
5	Female	Married	Guinea	U.S.	U.S.
6	Male	Married	Ethiopia	Ethiopia/2004	Ethiopia
7	Female	Divorced	Kenya	Kenya/1994	Kenya
8	Male	Single	Ethiopia	U.S./2013	U.S.
9	Female	Married	Kenya	Kenya/2005	Kenya
10	Female	Divorced	Kenya	Kenya/2005	Kenya
11	Male	Single	Ethiopia	U.S./2005	U.S.
12	Male	Single	Liberia	U.S./2009	U.S.
13	Female	Divorced	Kenya	Kenya	Kenya
14	Male	Married	Togo	U.S./2001	U.S.

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
