# Peer review of "The Significance of Taking Antiretroviral Medications for African-Born People Living with HIV and Residing in Minnesota"

_pharmacy, 2020, doi:10.3390/pharmacy8020108_

Round 1

Reviewer 1 Report

Dear author,

The manuscript touches a sensitive topic. 

I recommend a few modification.

Double check the English.

Authors should give more details about the period when the stady was performed, the number of participants in total. Also the methodology part lacks some details about the inclusions and exclusions criterias.

The tests that were used are not clear defined, mayba if the authors attached them to the manuscript, then the thing would be clearer for the reader.

To attribute significance to the results, qualitative study should be expressed in at least some objective manner and provide some statistics to support significance. Since the sample is very small, it can be expressed either in tables or charts as descriptive study results or case series.

Conclusion need to be short and clear. Please phrase 3 short conclusion.

Please recheck the References order.

Thank you!

Best regards!

Author Response

Point 1: Thank you for your time in reviewing our manuscript. We amended the document with more information about the inclusion and exclusion criteria and the total number of participants (14) for this study. The study recruited the participants until saturation was obtained. Once the saturation of the data occurred the study stopped recruiting the participants. Therefore, the answer is 14 participants. We amended the text to reflect this critical information. The main exclusion criteria were participants who did not speak English, were not born in Africa, and those who did not take antiretroviral therapy at the time of the interview despite being HIV positive.

Point 2: Thank you for asking for the methodology used in presenting the results. It has to be pointed out that qualitative research does not use statistical tests for data analysis. On the contrary, Thematic Analysis, as described by Braun and Clarke, was used for analysis.1,2

Point 3: Thank you for your suggestion. As stated above, qualitative analyses cannot use statistical methodology. Therefore, we used Braun and Clarke recommendations in presenting the data as quotes. Furthermore, this study used narrative inquiry, which highly recommends showing the corpus of data as themes with descriptions that further elucidate the message conveyed by the quotes. In light of reviewer’s three recommendations, we decided to use reviewer’s suggestions and keep the results.

Point 4: Thank you for this thoughtful recommendation. In light of the comments from reviewers 2, and 3, the conclusion section was amended. We addressed this suggestion by removing some of the details while also addressing the other reviewers’ suggestions.

Point 5: Done.

References

  1. Braun V, Clarke V. Using thematic analysis in psychology. Qualitative research in psychology. 2006;3(2):77-101.
  2. Braun V, Clarke V. Successful qualitative research: A practical guide for beginners. sage; 2013.

Reviewer 2 Report

38: Global new infections- please mention period of incidence after the number of infections.

39-40: please compare same year of global and US or Minnesota incidence, if possible.

Line 50: is it a rate in percent? What is the denominator?

Objectives in introduction differ from abstract. Please recheck.

Methods: Discuss how many participants in total, How they were enrolled? If your exposure is ART or HIV, outcome is quality of life tested by your themes, please define the exposure and outcome and how did you test them?

Quotes can be provided in appendix, please provide all questionnaires of themes in appendix.

Lot of subjectivity for each participant in discussion of the results. This subjectivity should be reduced, should identify what is the theme or barrier affecting the participants, should present it in tables. To attribute significance to the results, qualitative study should be expressed in at least some objective manner and provide some statistics to support significance.

This should be ideally an interventional study design. Intervention is ART, and authors are assessing quality of life before and after ART. The study design and results are not representing this. Even if the authors are not doing the intervention by themselves, they can use their themes of questionnaires to assess quality of life before and after ART and should compare the different results. The themes can be of narrative style, but the methods and results should not be of narrative style, if any significance should be given to the results.

Significance of the results needed in a table. Since the sample is very small, it can be expressed either in tables or charts as descriptive study results or case series.

The power of the sample is very less, since only 14 participants. If there is significance difference in study’s results, it cannot be identified by small sample. So, the results can be expressed in descriptive terms and tables.

Author Response

Point 1: Thank you for this suggestion. However, we can't compare the same year of global and US or Minnesota incidence. The most recent report from the Minnesota Department of Health discussed the rates of HIV infection by comparing different ethnicities. Consequently, we used this report to bring more evidence to our study and showed how the African-born PLWH are disproportionately affected by HIV in Minnesota.

Point 2: Thank you for this thoughtful recommendation to describe the denominator for the rate of the infection. Therefore, we amended the text to reflect these changes, and we believe that this addition strengthens our manuscript.

Point 3: Thank you for this thoughtful recommendation. We double checked and amended the text to have identical objectives in the abstract as in the methods.

Point 4: Thank you for this suggestion. We amended the document with more information about the inclusion and exclusion criteria and the total number of participants (14) for this study. The study recruited the participants until saturation was obtained. Once the saturation of the data occurred the study stopped recruiting the participants. Therefore, the answer is 14 participants. We amended the text to reflect this critical information. The main exclusion criteria were participants who did not speak English, were not born in Africa, and were HIV positive, but did not take antiretroviral therapy at the time of the interview. It has to be pointed out that qualitative research does not use statistical tests for data analysis. On the contrary, Thematic Analysis, as described by Braun and Clarke was used for analysis.1,2 Therefore, we used Braun and Clarke recommendations in presenting the data as quotes.1 Thank you for suggesting adding the interview questions that resonate with the extracted themes as an appendix. The manuscript was amended.

Point 6: Thank you for suggesting bringing more evidence to a qualitative study by using a table and further statistics. However, it would be erroneous to use statistics in the qualitative data presentation. Furthermore, this study used narrative inquiry, where it is highly recommended to show the corpus of data as themes with descriptions that tease out the message conveyed by the quotes.  In addition to utilizing the researchers’ extensive experience in the qualitative field, we also consulted various qualitative articles and books that recommend against describing the qualitative data using statistical information. In light of the reviewer’s 3 recommendations, we decided to use reviewer’s 3 suggestions and keep the results.

Point 6: Thank you for suggesting “to use their themes of questionnaires to assess quality of life before and after ART and should compare the different results.”

However, we believe it is not the scope of this study to compare the quality of life before and after ART. Since all the participants were initiated on ART regimen in different periods of their lives, this task would be impossible. Furthermore, this study aims to describe the medication experience from the participants’ point of view and not the quality of life.

Point 7: Thank you for suggesting presenting the results as a table or chart. However, as mentioned above, the qualitative methodology used by the study cannot be displayed as charts. It could be presented as a table in certain circumstances; the data was obtained from a survey which consisted of open-ended questions or a mixed-methods study. However, since this study used Narrative Inquiry as suggested by Braun and Clarke, the thematic analysis must be presented in a narrative style.

Point 8: Thank you asking about the power of the study. We amended the text and explained that qualitative studies do not use power for data analysis. On the contrary, qualitative studies do not rely on large sample sizes to claim validity for the concepts generated. Sample size in qualitative studies has a different meaning than in quantitative studies and does not rely on power. A target sample size for a qualitative study is ten participants or until detailed descriptions and rich themes emerge from the interviews, therefore reaching a level of saturation. Saturation has been defined as a point beyond which no significantly new information is obtained.

Therefore, we cannot present the data as tables.

References

  1. Braun V, Clarke V. Using thematic analysis in psychology. Qualitative research in psychology. 2006;3(2):77-101.
  2. Braun V, Clarke V. Successful qualitative research: A practical guide for beginners. sage; 2013.

Reviewer 3 Report

A well-written manuscript that describes the themes of African-born patients living with HIV in Minnesota. Definitely interesting to see how much spirituality played in these patients' lives and the impact that culture has on our patients. 

  • The abstract mentions 2 objectives for the study - however, the body of the manuscript really only addresses the first of capturing the experiences, would recommend removing 2nd objective or expanding on how you assess the impact of economic factors. 
  • Introduction is well written and provides a good overview of the problem while still being concise. 
  • Methods - Would be nice to see the interview questions either added as a table or supplementary document
  • Results - Again, nothing is results related to the 2nd objective stated in Abstract & Introduction
  • Discussion - Lines 340-342s & Lines 356-359 these are very short paragraphs (2 sentences) that could be expanded, add in the literature to show why these results are unique and important. 
  • Conclusion - well written, addresses objective 1 but not objective 2 of the study. 

Table 1. Is there a reason for Patient 12 not to have listed the country where treatment was started? 

Author Response

Point 1: We are grateful for asking this clarification. We went to the original manuscript and removed the economic considerations.

Point 2: Methods: Thank you for this recommendation. We amended the manuscript, by adding the interview questions related to these themes. The questions were added as an appendix.

Point 3: Results: We amended the text to reflect these changes.

Point 4: Discussion: Thank you for this suggestion. We amended the text to show why these results are unique and important.

Point 6: Conclusion: Thank you for this suggestion. We amended the text.

Point 7: Table 1. We are grateful for pointing out this mistake. We amended the table.

Round 2

Reviewer 2 Report

Results: In themes it was mentioned as 'some participants' or 'several participants'. If possible please provide the number of the participants applicable to those instances.

Strengths and limitations: If the study participants are immigrants from African countries, the study results can be extended to other immigrants from similar countries, since contexts can be similar. But the analysis findings may not be applicable to either people born and living in US and also to the people living in home countries, since social and environmental conditions and consequent contexts change. Please comment in the limitations.

Author Response

Point 1: Thank you for asking for this clarification. It is unusual for Narrative Inquiry interviews to present the number of participants who responded to some of the questions. Per your request, we amended the text.

Point 2: Thank you for this recommendation. It is well known that qualitative research is not generalizable. We wanted to avoid using this cliché in the limitations and focused on other limitations of the study. We amended the text to reflect it.